# Robust and Fragile Majorana Bound States in Proximitized Topological Insulator Nanoribbons

**DOI:** 10.3390/nano13040723

**Published:** 2023-02-14

**Authors:** Dennis Heffels, Declan Burke, Malcolm R. Connolly, Peter Schüffelgen, Detlev Grützmacher, Kristof Moors

**Affiliations:** 1Peter Grünberg Institute 9, Forschungszentrum Jülich & JARA Jülich-Aachen Research Alliance, 52425 Jülich, Germany; 2JARA-Institute for Green IT, RWTH Aachen University, 52056 Aachen, Germany; 3Blackett Laboratory, Imperial College London, South Kensington Campus, London SW7 2AZ, UK

**Keywords:** topological insulators, nanowires, Majorana, proximity-induced superconductivity, tunneling spectroscopy

## Abstract

Topological insulator (TI) nanoribbons with proximity-induced superconductivity are a promising platform for Majorana bound states (MBSs). In this work, we consider a detailed modeling approach for a TI nanoribbon in contact with a superconductor via its top surface, which induces a superconducting gap in its surface-state spectrum. The system displays a rich phase diagram with different numbers of end-localized MBSs as a function of chemical potential and magnetic flux piercing the cross section of the ribbon. These MBSs can be *robust* or *fragile* upon consideration of electrostatic disorder. We simulate a tunneling spectroscopy setup to probe the different topological phases of top-proximitized TI nanoribbons. Our simulation results indicate that a *top-proximitized* TI nanoribbon is ideally suited for realizing fully gapped topological superconductivity, in particular when the Fermi level is pinned near the Dirac point. In this regime, the setup yields a single pair of MBSs, well separated at opposite ends of the proximitized ribbon, which gives rise to a robust quantized zero-bias conductance peak.

## 1. Introduction

Three-dimensional (3D) topological insulators (TIs) have received a lot of attention in the last decade due to their interesting electronic properties, in particular due to their topologically protected surface states with a spin-momentum-locked Dirac-cone energy spectrum [1]. This interest has only increased when the possibility emerged to realize exotic forms of superconductivity by combining TIs and ordinary *s*-wave superconductors in heterostructures [2,3,4,5,6,7,8], exploiting the superconducting proximity effect [9]. Due to strong spin-orbit coupling in the TI, the induced superconductivity transforms into *p*-wave pairing for the TI surface states [10].

A promising application of *p*-wave superconductivity is to realize Majorana bound states (MBSs) in a spinless fermionic channel [11], forming a so-called Majorana wire. These MBSs come in pairs of states at zero energy, which are localized at opposite ends of the wire. A pair of MBSs can also be understood as a pair of quasiparticles forming equal-weight superpositions of a particle and hole state. As such, the MBS is a quasiparticle that is its own anti(quasi)particle, with an associated creation/annihilation operator that is self-adjoint [12]. Furthermore, these MBSs are anyons with non-Abelian exchange statistics [13]. By combining these exotic properties, MBSs can be exploited for quantum information processing with the promise of being immune against the most common sources of decoherence [14,15,16,17,18].

With the abovementioned properties, a superconductor-TI nanowire or nanoribbon heterostructure appears to be a natural Majorana wire candidate. Unfortunately, the surface-state spectrum suffers from a spin degeneracy, which naturally arises due to confinement quantization of the spin-momentum-locked Dirac cone spectrum with antiperiodic boundary conditions for the Dirac spinor solutions [19]. This prevents the realization of topological *p*-wave superconductivity with a single spinless channel that is underlying the realization of a Majorana wire. The unwanted degeneracy can be lifted, however, by applying an external magnetic field along the wire [19,20,21,22]. The magnetic flux through the cross section of the TI nanowire modifies the boundary condition for the surface states and thereby lifts the degeneracy that prevents the formation of a topologically nontrivial regime [23,24].

Even when the proximitized TI nanowire is brought into the topological regime with an external magnetic field, there is no guarantee that well-separated MBSs form at opposite ends of the wire. For this, a sizeable proximity-induced superconducting gap must be induced in the surface-state spectrum of the TI nanowire. In recent works, the realization of such a fully gapped nontrivial regime was brought into question, as it was found that the particle and hole states fail to couple due to a mismatch in transverse momentum and, hence, the proximity effect fails to induce a superconducting gap [25]. To overcome this mismatch, it has been proposed to consider a vortex in the superconducting condensate that envelopes the TI wire [25], or to break the transverse symmetry of the wire with an electric field by introducing an electrostatic gate in the device layout [26].

In this article, we scrutinize the conditions for fully gapped topological superconductivity in a proximitized TI nanoribbon (i.e., a nanowire with rectangular cross section) structure, considering a realistic device layout that is compatible with state-of-the-art nanofabrication processes [6]. In particular, we consider a selectively grown TI nanoribbon that is covered from the top by a conventional superconductor (see Figure 1a), e.g., Nb [27]. Through careful consideration of the proximity effect, we find that this setup naturally yields optimal conditions for fully gapped topological superconductivity when the ribbon cross section is pierced by (close to) half a magnetic flux quantum, with a single pair of *robust* MBSs appearing at the ends of the nanoribbon. For other flux values, we identify different gapless and gapped phases with different numbers of end-localized MBSs, depending on the position of the Fermi level with respect to the Dirac point. Some of these MBSs appear to be *fragile* when disorder is introduced in the system and suffer from hybridization with other MBS solutions on the same end of the ribbon. We also consider a tunneling spectroscopy setup for distinguishing robust and fragile MBSs and identifying these different phases in experiments.

Starting with this introduction, the article is divided into six sections. In Section 2, we cover the details of our simulation approach, including the continuum model Hamiltonian (Section 2.1), the treatment of the superconducting proximity effect (Section 2.2), and the tight-binding modeling approach (Section 2.3). In Section 3, we discuss the (proximity-induced) spectral gap and the topologically trivial and nontrivial regimes of a top-proximitized TI nanoribbon. In Section 4, we discuss the different phases of this system, with different numbers of robust and fragile MBSs, which can be probed with tunneling spectroscopy (Section 4.1). We proceed with a discussion of the simulation results in Section 5 and conclude in Section 6.

## 2. Model

### 2.1. Topological Insulator

For obtaining the TI nanoribbon spectrum, we consider a tight-binding model (see Section 2.3) that is derived from the following 4-band continuum model Hamiltonian for the Bi_2_Se_3_ family of TI materials [28,29]:(1)$$\begin{array}{cc} H_{0}\left(k\right) & =\left[\epsilon \left(k\right)-\mu \right]+M\left(k\right)\sigma_{z}+A_{⊥}\left(k_{y}s_{x}-k_{x}s_{y}\right)\sigma_{x}+A_{z}k_{z}\sigma_{y}\\ \epsilon (k) & \equiv C_{0}-C_{⊥}\left(k_{x}^{2}+k_{y}^{2}\right)-C_{z}k_{z}^{2},\;M\left(k\right)\equiv M_{0}-M_{⊥}\left(k_{x}^{2}+k_{y}^{2}\right)-M_{z}k_{z}^{2}. \end{array}$$
$H_{0}\left(k\right)$ is a 4×4 matrix that is written as a linear combination of a tensor product of Pauli matrices $s_{a}$ and $\sigma_{b}$ (a,b∈{x,y,z}) acting on the spin and orbital (pseudospin) subspaces, respectively. Note that the identity matrices are not written explicitly here. The parameters $C_{0},C_{⊥},C_{z},M_{0},M_{⊥},M_{z},A_{⊥},A_{z}$ can be obtained for different TI materials, such as Bi_2_Se_3_ or Bi_2_Te_3_ [29]. Here, we will neglect in-plane (⊥) versus out-of-plane (*z*) anisotropy (in terms of model parameters, $A_{⊥}=A_{z}\equiv A,C_{⊥}=C_{z}\equiv C,M_{⊥}=M_{z}\equiv M$) and asymmetry between valence and conduction bands (C=0) for simplicity (these simplifications will not significantly affect the surface-state spectrum near the Dirac point [30], which is the focus of this work). We consider remaining model parameters A=3eV·Å, $M=15\mathrm{eV}\;\cdot \;{Å}^{2}$, and $M_{0}\mathrm{eV}=0.3$ to represent a 3D TI material with inverted (direct) bulk band gap at the Γ point equal to 0.6eV and a Dirac velocity for the 3D TI surface states equal to $v_{D}=A/\hbar \approx 4.6\times 10^{5}\;m/s$, which are both comparable to Bi_2_Se_3_ [28] (note that 0.6eV reflects the band separation at the Γ point and not the overall bulk band gap, which is closer to 0.3eV).

For describing a charge carrier density fluctuations in the ribbon (i.e., electrostatic disorder), we can add a disorder term $S_{\mathrm{dis}}\phi \left(r\right)$ to the parameter $C_{0}$ ($\mu =C_{0}$ corresponds to the chemical potential being pinned at the Dirac point in a pristine system without disorder). This disorder is characterized by a fluctuation amplitude $S_{\mathrm{dis}}$ and the function ϕ(r), which we consider to be a unit-normalized white-noise profile [$\langle \phi \left(r\right)\phi \left(r^{\prime }\right)\rangle =\delta \left(r-r^{\prime }\right)$] or a Gaussian random field [$\langle \phi \left(r\right)\phi \left(r^{\prime }\right)\rangle =e^{-{\left(r-r^{\prime }\right)}^{2}/\left(2\lambda^{2}\right)}$] with spatial correlation length λ. Here, we consider an electrostatic disorder strength of the order of the TI surface-state subband spacing (∼2πA/P, see below, which is in the 10meV range) and a spatial correlation length in the few-nm range [31].

### 2.2. Proximity-Induced Superconductivity

In our setup, we consider a 3D TI nanoribbon that is proximitized via its top surface by an *s*-wave superconductor. We make use of the Bogoliubov–de Gennes (BdG) formalism to treat the proximity-induced superconducting pairing [32], which yields the following model Hamiltonian:(2)$$H=\frac{1}{2}\sum_{k}\Psi_{k}^{†}\underset{=H_{\mathrm{BdG}}}{\underset{︸}{\left(\begin{array}{cc} H_{0}\left(k\right) & \underline{\underline{\Delta }}\\ {\underline{\underline{\Delta }}}^{†} & -H_{0}^{*}\left(-k\right) \end{array}\right)}}\Psi_{k},$$
with the Nambu spinor $\Psi_{k}^{†}=\left(c_{k}^{†},c_{-k}^{T}\right)$ with $c_{k}^{†}=\left(c_{kA↑}^{†},c_{kB↑}^{†},c_{kA↓}^{†},c_{kB↓}^{†}\right)$. Here, $c_{k\alpha }^{†}$ ($c_{k\alpha }$) are creation (annihilation) operators that form a basis for $H_{0}$ ($-H_{0}^{*}$) with degrees of freedom α,β∈{↑A,↑B,↓A,↓B}.

We consider conventional *s*-wave pairing (induced by the superconductor on top of the TI nanoribbon) in the BdG formalism, which is given by a momentum-independent pairing
(3)$$\underline{\underline{\Delta }}=iℜ\left\{\Delta \right\}s_{y}-ℑ\left\{\Delta \right\}s_{y},$$
with complex-valued pairing potential Δ=ℜ{Δ}+iℑ{Δ}. Combining Equations (Equation 1)–(Equation 3), we obtain the following 8-by-8 BdG-Hamiltonian matrix:(4)HBdG(k)=[ϵ(k)−μ]τz+M(k)τzσz+A⊥(kysx−kxτzsy)σx+Azkzτzσy=−ℜ{Δ}τysy−iℑ{Δ}τxsy,
with $\tau_{c}$ (c∈{x,y,z}) Pauli matrices acting on the particle–hole subspace.

Note that the superconducting proximity effect encompasses several aspects, such as an induced superconducting gap, an induced pairing potential, and induced particle–hole correlations [33]. For our simulation approach, based on the BdG Hamiltonian in Equation (Equation 4), we do not explicitly include the superconductor on top and only consider the pairing potential that is induced at the interface (see Ref. [34] for an explicit derivation of such a pairing term at the interface) as input. With this input, we can calculate the surface-state quasiparticle spectrum at low energies, with induced spectral (superconducting) gap, as well as particle–hole correlations throughout the complete TI nanoribbon, for example [35].

### 2.3. Tight-Binding Model

For simulating a proximitized TI nanoribbon, we discretize the BdG Hamiltonian of Equation (Equation 4) via the standard procedure onto a regular cubic grid with lattice constant a=1nm, resulting in a tight-binding model Hamiltonian with on-site and nearest-neighbor hopping matrices, $H_{\mathrm{onsite}}$ and $H_{\mathrm{hop}\;x,y,z}$, respectively.

For the BdG-Hamiltonian $H_{\mathrm{BdG}}\left(k\right)$ described in Equation (Equation 4), this results in the following matrices:(5)Honsite=C0−4C⊥+2Cza2−μτz+M0−4M⊥+2Mza2τzσz,=−ℜ{Δ}τysy−iℑ{Δ}τxsyHhopx=C⊥a2τz+M⊥a2τzσz+iA⊥2aτzsyσx,
with comparable hopping matrices along the *y* and *z* directions.

All the simulation results presented in this work are obtained with this tight-binding model (see Appendix A for details on the implementation of the simulation approach and retrieval of the spectral gap). For our setup, we consider a nanoribbon with infinite length *L* or L=1μm along the *x*-direction, and a square cross section (width W=10nm along *y*, height H=10nm along *z*, perimeter P=2W+2H=40nm) that is proximitized by an *s*-wave superconductor covering its top surface (see Figure 1a). Because the TI is not an intrinsic superconductor, the pairing potential decays quickly away from the TI-superconductor interface, i.e., over atomic distances [35]. It is therefore reasonable to assume a nonzero Δ (here, we consider Δ to be real, without loss of generality, and equal to 5meV) only on the topmost layer of the TI nanoribbon lattice model (∼1nm thick) that is considered to be in direct contact with the superconductor, while Δ=0 elsewhere in the lattice.

For the orbital effect of an external magnetic field, we consider the Peierls substitution method for the hopping terms:(6)$$t_{i\to j}\to t_{i\to j}\exp\left(i\frac{q}{\hbar }\int_{r_{i}}^{r_{j}}dr\;\cdot \;A\left(r\right)\right),$$

Here, $t_{i\to j}$ represents the hopping matrix from site *i* with position $r_{i}$ to *j* with position $r_{j}$, which is modified by Peierls substitution with a phase depending on the vector potential A with corresponding external magnetic field $B=\vec{\nabla }\times A$, reduced Planck’s constant *ℏ*, and the charge q=∓e (with *e* the elementary charge) of the charge carrier (either a particle or a hole). We consider a constant external magnetic field oriented along the nanoribbon (in the frame of reference considered here, along the *x*-direction) with A=(0,|B|(z−H),0). We consider this vector potential such that it vanishes on the topmost layer of the nanoribbon (z=H) and is compatible with A=B=0 for z>H, avoiding any supercurrent ∝|Δ|A in our description [32]. These assumptions correspond to a simplified treatment of the experimental setup: a vector magnetic field that vanishes completely inside the superconductor on top of the TI nanoribbon while neglecting any shielding current.

## 3. Spectral Gap

In Figure 1c, we present the gap in the quasiparticle spectrum of a top-proximitized TI nanoribbon as a function of magnetic flux piercing the cross section of the ribbon and of chemical potential (note that μ=0 corresponds to the position of the Dirac point of the TI surface-state Dirac cone). The parameter space is divided into different gapped regions that are separated by gapless phase boundaries or regions. In general, the gap lies somewhere between zero and |Δ|, with Δ the superconducting pairing potential considered on the top surface of the TI nanoribbon (in direct contact with the proximitizing superconductor), which provides a natural upper bound for the proximity-induced superconducting gap. Only near μ=0 and integer multiples of the flux quantum does the gap exceed |Δ|. Here, the evaluated spectral gap is a trivial insulating gap due to confinement quantization ∼πA/P, rather than a proximity-induced superconducting gap.

The spectral gap is either topologically trivial or nontrivial, and the top-proximitized TI nanoribbon only forms a quantum wire with unpaired MBSs in the nontrivial regime. The nature of the gap can be determined with the following $Z_{2}$ topological invariant [11],
(7)$$\begin{array}{c} M=\mathrm{sign}\{\mathrm{Pf}\left[H_{\mathrm{TINR}}\left(k=0\right)\right]\mathrm{Pf}\left[H_{\mathrm{TINR}}\left(k=\pi /a\right)\right]\}, \end{array}$$
with Pf short for the Pfaffian and $H_{\mathrm{TINR}}\left(k\right)$ the tight-binding model Hamiltonian over the cross section of the TI nanoribbon with wave number *k* along the direction of the ribbon. The trivial and nontrivial regions are indicated by color and are in good qualitative agreement with the diamond-tiled phase diagram that can be obtained analytically for a cylindrical TI nanowire model with Δ=0 [23]. A (nonproximitized) cylindrical TI nanowire has the following surface-state (particle) spectrum:(8)$$\begin{array}{c} E_{l}\left(k\right)=\pm \hbar v_{D}\sqrt{k^{2}+{\left(2\pi l+1/2-\eta \right)}^{2}/P^{2}}-\mu , \end{array}$$
with *k* the wave number (for propagation along the nanowire), l=0,±1,… the quantum number for quantized angular momentum, $v_{D}$ the Dirac velocity of the Dirac cone, $\eta =\Phi /\Phi_{0}$ the total magnetic flux piercing the nanoribbon cross section in multiples of flux quanta ($\Phi_{0}\equiv h/e$), and P=2πR the diameter of the cylindrical nanowire with radius *R*. From this expression, it can be seen that the spectrum is a subband-quantized Dirac cone that is flux quantum-periodic. By evaluating the number of Fermi points ν with k>0 (or k<0), i.e., the number of forward (or backward)-propagating surface states at zero energy from the different subbands that cross the chemical potential, the topological invariant above can be obtained in an alternative way by evaluating $M={\left(-1\right)}^{\nu }$. In other words, the system becomes topologically nontrivial when there is an odd number of such Fermi points and corresponding propagating modes. Each diamond in the phase diagram represents a bounded region with a given number of Fermi points, which is even for the diamonds in grayscale, and odd for the diamonds in redscale. Hence, for a piercing magnetic flux that is a half-integer multiple of one flux quantum, the system always has an odd number of Fermi points and remains in the nontrivial regime for all values of the chemical potential μ (within the TI nanoribbon bulk gap, as we are only considering the topological surface states inside the bulk gap). Conversely, the system is always in the trivial regime without an external magnetic field, or when the piercing magnetic flux is an integer multiple of $\Phi_{0}$.

Note that Fermi points, i.e., zero-energy surface states, can only be considered in general for Δ=0, as the states can otherwise gap out around zero energy. Therefore, we resort to the more general topological invariant in Equation (Equation 7) and the perfect diamond tiling gets slightly deformed, with different diamonds in the phase diagram becoming connected. Further note that the diamond height (its extent as a function of chemical potential) is equal to the subband energy spacing and smaller than 2πA/P (with $A=\hbar v_{D}$), which is the expected spacing, based on the cylindrical wire model when substituting the diameter with the perimeter of the square cross section. This can also be seen in the surface-state spectrum presented in Figure 2a and has been reported before [30]. It may originate from the pile-up of wave function density near the corners, which appears to increase the *effective* perimeter of the cross section.

An interesting finding is that the spectral gap in the nontrivial region is maximal for the diamond closest to the Dirac point (μ=0), which corresponds to the region with a single Fermi point, and remains large for the diamonds at higher μ for a piercing flux close to a half-integer flux quantum (see Figure 1d). The size of this spectral gap agrees well with the perturbative estimate $E_{\mathrm{gap}}\approx \langle \psi_{p}|\underline{\underline{\Delta }}|\psi_{h}\rangle ∼\left(W/P\right)\Delta \approx \Delta /4$ for our TI nanoribbon with square cross section, with $|\psi_{p}\rangle$ and $|\psi_{h}\rangle$ particle and hole surface states at the Fermi point with Δ=0. The reduction factor 1/4 originates from the ratio of the section of the perimeter with nonzero Δ (only the top surface) to the complete perimeter, which gets enveloped by the particle and hole surface states.

In addition to the separation into trivial (grayscale) and nontrivial (redscale) regions, the spectral gap also reveals different gapped phases within the trivial and nontrivial regions themselves, separated by gapless phase boundaries (note that gap closings without a change of topological invariant were also reported in Ref. [34]). This suggests that the regions with and without unpaired MBSs subdivide further into different phases with additional distinct properties. As some of the regions stretch out over multiple diamonds, we can already rule out that the properties are strictly related to the number of Fermi points when Δ=0.

To reveal the properties of the different phases in the phase diagram, based on the spectral gap, we take a closer look at four points (indicated by ■, ★, •, and ▲), which lie in different diamonds or regions separated by a gapless boundary in Figure 1c. Their quasiparticle spectrum is presented in Figure 2b. In general, we see the expected number of Fermi points of the different subbands, based on the diamond to which the point belongs (a single Fermi point in the diamond of ■, two Fermi points in the diamond of •, and three Fermi points in the diamond of ★ and ▲), and local minima of the proximity-induced superconducting quasiparticle gap forming near them. Interestingly, ★ belongs to the same phase that stretches out over the complete *single-Fermi point* diamond below, in which ■ also lies, while ▲ sits in the same diamond and represents a different phase, separated from ★ by a gapless boundary. The qualitative difference between the two phases in the quasiparticle can be narrowed down to the number of local minima that can be attributed to a single spinless channel. In the diamond with three Fermi points where ★ and ▲ lie, there is either a single such local minimum or three of them, depending on the relative positioning of the Fermi points in reciprocal space. For ★, two Fermi points overlap in momentum, which effectively turns these channels into a trivial spinful channel. Hence, the number of spinless channels is reduced to one, which is the same numbers as in the diamond below. This spectral property has consequences for the formation of MBSs, as will be discussed in the following section.

## 4. Robust and Fragile Majorana Bound States

In Figure 3a, we present the low-energy quasiparticle spectrum as a function of flux of a top-proximitized TI nanoribbon with finite length. In this way, we also reveal the states that are localized at the ends of the ribbon. We fix the chemical potential to two different values to explore the different trivial and nontrivial phases, as discussed in the section above. Near $\Phi =n\Phi_{0}$ (n∈Z), there is a completely trivial insulating phase without any subgap states. For other values of the flux, however, subgap states appear in the spectrum, even in regions that are trivial according to the topological invariant in Equation (Equation 7), and they are localized at both ends of the nanoribbon (see Figure 4a).

Close to $\Phi =\left(n+1/2\right)\Phi_{0}$ and for the complete nontrivial diamonds nearest to the Dirac point (the connected region containing ■ and ★, for example), there are only two subgap states. These can be identified as a pair of MBSs forming at opposite ends of the top-proximitized TI nanoribbon. In other words, this phase corresponds to the conventional Majorana quantum wire system. This phase and its MBSs are robust against local disorder, as can be seen in Figure 3b,c, where it is presented how the low-energy spectra of Figure 3a are affected by increasing electrostatic disorder throughout the nanoribbon.

In the region with •, four subgap states form, two on each end of the ribbon. In this case, the TI nanoribbon effectively has two independent spinless channels (see previous section and Figure 2), which both give rise to the formation of a pair of MBSs forming at opposite ends of the nanoribbon. However, as there are two MBSs on each wire end, these MBSs can couple to the other MBS on the same end when they are exposed to local electrostatic disorder. Hence, in the presence of disorder, these MBSs hybridize into non-self charge-conjugate bound states with finite energy. We refer to such MBSs as *fragile* MBSs. In contrast, a single unpaired MBS can only suffer from hybridization with the MBS on the opposite end of the ribbon, which can be suppressed by making the proximitized section significantly longer than the MBS localization length $∼\hbar v_{D}/E_{\mathrm{gap}}$. Therefore, we refer to it as a *robust* MBS pair. The low-energy spectrum of ▲ shows six subgap states, which can be identified as three pairs of MBSs that form at opposite ends of the TI nanoribbon. When electrostatic disorder is present, two out of the three MBSs can hybridize locally and are thus fragile, while a single robust pair should remain protected as long as the TI nanoribbon remains in the nontrivial regime (similar to what happens in a tri-junction of Majorana wires [16]). When the electrostatic disorder strength becomes of the order of the diamond height ∼2πA/P, the TI nanoribbon is not guaranteed to remain in in the same phase when the flux is not an integer ($\Phi =n\Phi_{0}$) or half-integer [$\Phi =\left(n+1/2\right)\Phi_{0}$] multiple of the flux quantum. In this strongly disordered regime, the spectral gap will fluctuate strongly along the disordered nanoribbon and locally cross gapless phase boundaries. Because of this, many states with near-zero energies can appear, which renders it difficult to interpret the spectrum in terms of end-localized MBS pairs.

Now, we can put all these findings together with the phase diagram and spectral properties obtained in the previous section. It becomes clear that the different gapped phases can be identified by the number of MBS pairs forming at opposite ends of the proximitized TI nanoribbon, with the number always being even (odd) in the trivial (nontrivial) regime. When there is some amount of electrostatic disorder in the proximitized TI nanoribbon, however, it is expected that any even number of MBSs will hybridize locally with the other MBSs at the same end, while a single pair of MBSs should remain immune from hybridization in a nontrivial regime with an odd number of pairs in total. This single pair will then survive near zero energy at opposite ends of the proximitized TI nanoribbon. We can thus identify and label the phases in Figure 1c (also see Figure 1b) by their number of robust MBS pairs (zero or one) and fragile MBS pairs (an even number). As the phase with a single robust MBS pair and zero fragile pairs stretches out over a large chemical potential window near $\Phi =\left(n+1/2\right)\Phi_{0}$, it is the most robust phase with respect to electrostatic disorder. It does not suffer from fragile MBS, nor from strong fluctuations of the spectral gap. In the subsection below, we discuss characteristic signatures of these different phases and their MBSs in a tunneling spectroscopy setup.

### 4.1. Tunneling Spectroscopy

In this subsection, we consider the tunneling conductance of a metallic tunneling probe that is attached to an uncovered end of a proximitized TI nanoribbon (see Appendix A for details), as a function of piercing magnetic flux and energy of the injected carriers from the tunneling probe (corresponding to bias voltage across the tunneling junction). Note that, in the experimental setup, the uncovered end should be shorter than the induced coherence length $∼\hbar v_{D}/E_{\mathrm{gap}}$ [9] for obtaining a *hard* proximity-induced gap and clearly revealing the subgap states. The results of these simulations are shown in Figure 3d–f for the same TI nanoribbon and parameters as in Figure 3a–c, and in Figure 4b for fixed values of the piercing magnetic flux. We consider conductance normalized to $G_{0}\equiv e^{2}/h$.

Overall, we find that the tunneling conductance reveals the subgap spectrum of the top-proximitized TI nanoribbon without disorder well, with a quantized conductance peak of $2e^{2}/h$ near zero bias (corresponding to perfect Andreev reflection) when a single unpaired MBS is localized on the side of the tunneling probe. Due to the finite length of the ribbon, there is hybridization of MBS pairs across the length of the ribbon, resulting in a splitting of the conductance peak away from zero bias. This splitting is modulated by the piercing magnetic flux.

When electrostatic disorder is introduced, the tunneling conductance of the phase with only a single (robust) MBS pair remains qualitatively the same. The zero-bias conductance peak is wider in the case of strong disorder, but it remains quantized and pinned at zero energy (see Figure 4b). The tunneling conductance of the phases with (fragile) MBS pairs gets heavily affected by disorder, however, and a quantized conductance peak cannot be easily identified. Instead, the conductance near zero bias displays irregular signatures that are very sensitive to the bias and piercing flux. This can be expected as the subgap spectrum itself is heavily affected by the disorder. Hence, it will be harder to identify phases with fragile MBSs via tunneling spectroscopy, and to determine whether the phase lies in the trivial or nontrivial regime, in particular when the electrostatic disorder strength is of the order of the subband spacing or larger.

## 5. Discussion

It is important to note here that the realization of fully gapped topological superconductivity in proximitized TI nanoribbons has been considered before with comparable simulation approaches [23,24,25,26,34,36]. It was pointed out by de Juan et al. in Ref. [25] that some form of transverse asymmetry is required to open up a superconducting gap in the TI nanoribbon surface-state quasiparticle spectrum. Without asymmetry, the induced gap is expected to vanish ($E_{\mathrm{gap}}\approx \langle \psi_{p}|\underline{\underline{\Delta }}|\psi_{h}\rangle \approx 0$) because of a mismatch of quantized transverse momentum. Transverse asymmetry can be induced by a superconducting vortex enveloping the nanoribbon [25], by electrostatic gating [26], or by considering more sophisticated hybrid structures with multiple superconductors and gates, which may also enhance the proximity-induced gap away from the Dirac point [34], for example. Our results, however, suggest that the strong decay of the pairing potential away from the intrinsic superconductor is already sufficient to realize the required transverse asymmetry when only bringing one of the side surfaces (e.g., the top surface) in direct contact with the superconductor. In this way, a sizeable proximity-induced gap, ∼(W/P)|Δ| or ∼(H/P)|Δ|, can be naturally achieved near a half-flux quantum of piercing magnetic flux, especially when the Fermi level is close to the Dirac point [34]. Hence, our findings suggest that a rather straightforward device layout (a TI nanoribbon onto which superconducting material is deposited) should already be ideally suited for realizing the topologically nontrivial regime with well-separated unpaired MBSs.

Furthermore, we note that multiple (fragile) MBSs are not unique to top-proximitized TI nanoribbons. They also appear in proximitized semiconductor nanowires with a multi-subband treatment [37], for example. In that case, however, the nontrivial regions with a different number of Fermi points are disconnected, such that electrostatic disorder can more easily push the system into a trivial regime [38].

For future work, the consideration of material and sample-specific parameters for the, e.g., TI model Hamiltonian, ribbon dimensions, and disorder strength would be interesting to explore. Equally relevant is the consideration of more complicated MBS architectures that allow for braiding, with multiple proximitized nanoribbons with different orientations as building blocks (e.g., a Y-junction [24]). For such structures, the orbital effect from an external magnetic field that is misaligned with one of the ribbons should also be considered. This has been shown to induce a steering effect in TI nanoribbon structures [30,39], and also affects the topological phase in semiconductor nanowires, for example [40,41,42,43].

Regarding the experimental feasibility, we expect that the phase diagram presented here can be resolved in state-of-the-art TI nanoribbon samples. Quasi-ballistic transport of topological surface states [44,45] (also in combination with tunability of the Fermi level with respect to the Dirac point via electrostatic gating [46,47,48,49,50]), surface-state subband quantization [51], as well as proximity-induced superconductivity [8], have all been demonstrated in TI nanowires or ribbons. Aside from electrostatic gating, heterostructure engineering can be considered to tune the intrinsic Fermi level within a few meV from the Dirac point [52,53].

Finally, we comment on the consideration of tunneling spectroscopy to probe the different topological phases of top-proximitized TI nanoribbons. From alternative MBS platforms (in particular, semiconductor nanowires), we know that a (quantized) zero-bias conductance peak as MBS signature must be considered with care, as such a signature can also have a trivial origin [54]. Nonetheless, the conditions for such false positives are less likely to appear in top-proximitized TI nanoribbons because of two important differences. First, topological TI surface states have more intrinsic robustness against disorder than the low-energy modes of semiconductor nanowires due to spin-momentum locking [55]. Second, a quantum dot at the end of the ribbon, which is one of the important mechanisms for retrieving a zero-bias conductance peak with trivial origin in semiconductor nanowires [54], is less likely to form because of the linear Dirac-cone spectrum.

## 6. Conclusions

With a detailed three-dimensional tight-binding model, we investigate numerically the spectral gap of three-dimensional topological insulator nanoribbons with magnetic flux-piercing and proximity-induced superconductivity, induced by a superconductor on the top surface. The spectral gap reveals a rich phase diagram as a function of flux and chemical potential, with different gapped phases with paired and unpaired Majorana bound states appearing at opposite ends of the nanoribbon. These Majorana bound states can be robust or fragile with respect to local hybridization due to electrostatic disorder. When the Fermi level in the topological insulator nanoribbon is close to the Dirac point and the piercing magnetic flux is close to half a flux quantum, we retrieve the optimal conditions for realizing fully gapped topological supperconductivity. With these conditions, there is a single pair of robust MBSs on opposite sides of the nanoribbon over an extended range of chemical potential and flux values. This phase gives rise to a robust quantized zero-bias conductance peak in tunneling spectroscopy.

## Figures and Tables

**Figure 1 nanomaterials-13-00723-f001:**
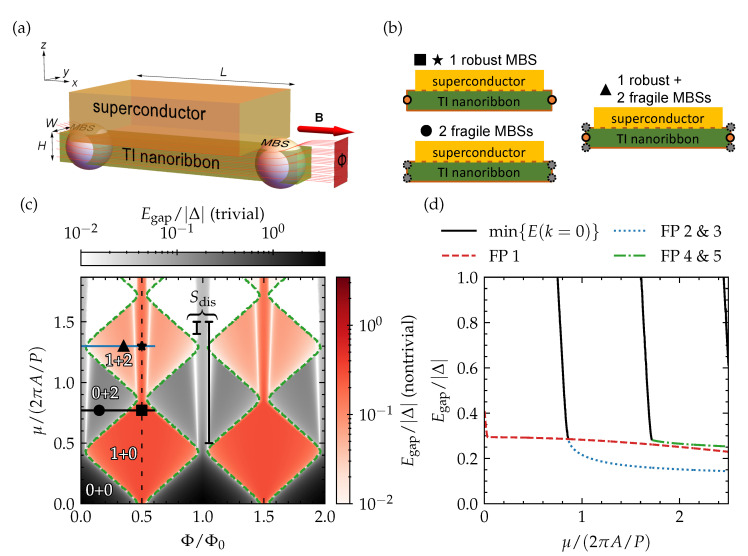
(**a**) A 3D TI nanoribbon that is proximitized by an *s*-wave superconductor via its top surface and pierced by a magnetic flux originating from an external magnetic field oriented along the ribbon. With this setup, MBSs form at opposite ends of the nanoribbon. The setup has different phases with different numbers of robust and fragile MBSs, which are shown schematically in (**b**); (**c**) the spectral gap in the proximitized TI nanoribbon as a function of magnetic flux and chemical potential. The grayscale (redscale) colormap indicates whether the gap is topologically trivial (nontrivial). The phase boundaries between the trivial and nontrivial regions are indicated with green dashed lines. The number of robust plus the number of fragile MBS pairs is indicated for some representative phases; (**d**) a line cut of the spectral gap shown in (**c**) as a function of chemical potential with magnetic flux equal to half a flux quantum (indicated by a vertical black dotted line in (**c**)). The gap of each surface-state subband that has a Fermi point (FP) when Δ=0 is presented separately, as well as the gap of the lowest subband without Fermi point, evaluated at k=0.

**Figure 2 nanomaterials-13-00723-f002:**
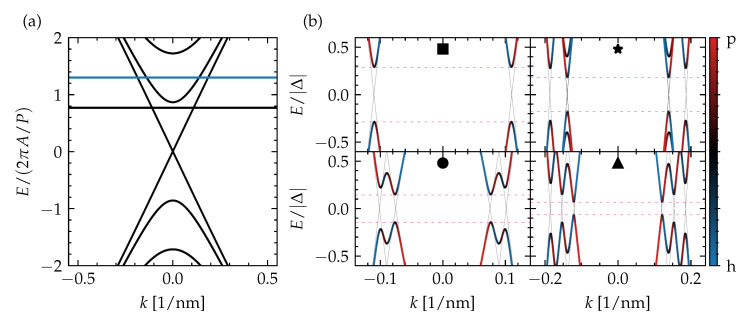
(**a**) The subband-quantized surface-state spectrum of a TI nanoribbon pierced by half a magnetic flux quantum. The two chemical potential values indicated with line cuts in Figure 1c and considered in (**b**) are indicated with horizontal lines; (**b**) the low-energy quasiparticle spectrum of a top-proximitized TI nanoribbon with infinite length for different combinations of piercing magnetic flux and chemical potential, as indicated in Figure 1c, with the color indicating the particle–hole (p-h) mixing (blue for hole, red for particle). The spectrum for Δ=0 is presented with thin black lines and the extent of the spectral gap is indicated with horizontal pink dashed lines.

**Figure 3 nanomaterials-13-00723-f003:**
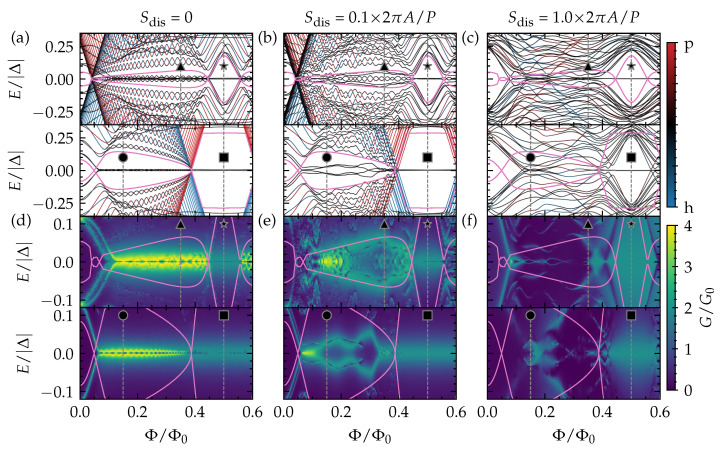
The (**a**–**c**) low-energy quasiparticle spectrum of a top-proximitized TI nanoribbon with finite length as a function of piercing magnetic flux Φ, and the (**d**–**f**) tunneling conductance as a function of piercing magnetic flux and energy *E* (or bias over the tunneling junction). The pink lines correspond to the spectral gap in the TI nanoribbon with infinite length and without disorder. The top (bottom) panel of each subfigure corresponds to the upper (lower) horizontal line cut shown in Figure 1c at μ≈1.3×2πA/P (μ≈0.75×2πA/P). The results are obtained (**a**,**d**) without disorder, and (**b**,**c**,**e**,**f**) with disorder (considering $S_{\mathrm{dis}}=0.1\times 2\pi A/P$ in (**b**,**e**), and $S_{\mathrm{dis}}=2\pi A/P$ in (**c**,**f**), with the different disorder strengths also indicated in Figure 1c).

**Figure 4 nanomaterials-13-00723-f004:**
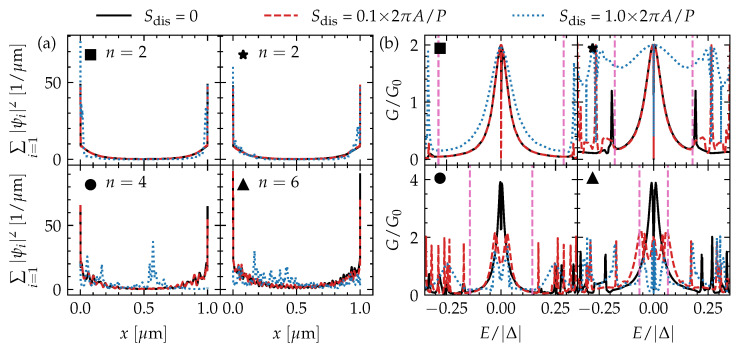
The (**a**) one-dimensional wave function density summed over the *n* lowest-energy (in absolute value) solutions and (**b**) tunneling conductance as a function of energy of a 1μm-long top-proximitized TI nanoribbon with and without disorder with different disorder strengths (indicated in Figure 1c) for four different combinations of chemical potential and piercing magnetic flux (also indicated in Figure 1c). The tunneling conductance in (**b**) corresponds to the vertical gray dashed line cuts in Figure 3d–f. The spectral gap of the TI nanoribbon considering infinite length and no disorder is indicated by vertical pink dashed lines.

## Data Availability

Data available in a publicly accessible repository.

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
