# Peer review of "Robust and Fragile Majorana Bound States in Proximitized Topological Insulator Nanoribbons"

_nanomaterials, 2023, doi:10.3390/nano13040723_

Round 1

Reviewer 1 Report

1st review of "Robust and fragile Majorana bound states in proximitized topological insulator nanoribbons" by Dennis Heffels et al.

The Authors study the Majorana bound states (MBS) in proximized TI nanoribbons. Papers sound interestingly, however, few modification before acceptance should be introduce.

A. Second paragraph should be more precisely written. In Kitaev model (ref. 11) the spinless fermions were studied in the p-wave (inter-site) type pairing.  In the spinfull fermion models, situation is more complicated. This paragraph should be improved.

B. The Authors study nanoribbons (more 2D than 1D system), but in the introduction are too small information about MBS in such systems -- more information should be added.

C. If TI model is correct, then in the presence of open boundary conditions (w/o SC), the Dirac cones should be observed. This is observed in calculations? How this states affect on the MBS?

D. The TI was define in momentum space (Eq. 1) , while influence of the flux is given by the Peierls substitution (Eq. 6). Relation between this two descriptions is not given directly in text, what can lead to some confiusions. For example, how the Authors find the spectrum of the system with open boundary condition?

E. Eq. 3 is not clear - there was assume complex SC order parameter? If there was no reason to this why Delta has this form when you tell about s-wave?

F. Fig. 1(a) - magnetix flux is perpendicular to the crossection of the nanoribbon. What was motivation to this assumption? What happend when B is nor directly parallel to the nanoribon?

G. Fig.2 band inversion is not well visible (color bar should be modified).

H. In some range of parameters two fragile MBS are observed - how and where are localized this states?

Aditionall comments:

Delta is not "superconducting paring potential" but superconducting order parameter (pairing potential will be U not described in this type models).

The MBS in the presence of orbital effects, was discuss also in J. Phys.: Condens. Matter 29, 495301 (2017), not mentioned in the paper.

Reviewer 2 Report

This manuscript describes the possible realization of Majorana bound states (MBSs) using a superconducting proximitized topological insulator (TI) nanoribbon. The authors systematically investigated the robustness of MBSs depending on the chemical potential and applied magnetic flux. Such platforms need to be suggested and developed in different shapes to accelerate experimental observation of MBSs. In this point, this paper can be published in Nanomaterials after a moderate revision considering the following questions. 

1) Sitthison and Stanescu have pointed out that a single-interface superconductor-TI nanoribbon structure, similar to the present study, is not a promising platform for realizing topological superconductivity [Phys. Rev. B 90, 035313 (2014)]. The authors need to address the difference in their model or calculation with comparison. 

2) The authors suggested a tunneling spectroscopy setup to probe the topological phase induced in the top-proximitized TI nanoribbons. However, it seems not experimentally feasible — to proximitize the top surface of a TI nanoribbon, a superconducting layer should be on the top, and the STM measurement cannot detect information underneath. 

3) They considered the thickness of the proximitized layer 1 nm, but it seems to be an improper assumption. One needs to consider the superconducting coherence length and the thickness of the surface state [Phys. Rev. Mater. 3, 124803 (2019)], which could be much longer than the one they assumed. Is there any alteration in their results if considering that? 

Reviewer 3 Report

The manuscript presents an theoretical study of Majorana bound states (MBSs) in topological insulator nanoribbons with proximity-induced superconductivity. The authors show that when the Fermi level is close to the Dirac point and the piercing magnetic flux is close to half of the flux quantum, a single pair of robust MBSs appears on opposite sides of the nanoribbon, leading to a robust quantized zero-bias conductance peak in tunneling spectroscopy. The presented results are of significant importance for the realization of robust MBSs, so the paper is worthy of publication in Nanomaterials.

I have two comments to section 2.1.

1)      On Page 2, lines 82-84, the authors make several simplifications to the Hamiltonian H0(k). It is not clear why these simplifications are needed and how they might affect the results presented in the manuscript.

2)      On page 3, lines 85-88, the authors write that the Hamiltonian H0(k) describes a 3D topological insulator with the bulk band gap of 0.6 eV, comparable to Bi2Se3. However, the bulk band gap of Bi2Se3 is about 0.3 eV (see Ref. [28]).

In conclusion, I recommend the manuscript for publication if the authors improve it taking into account the two comments above.

Round 2

Reviewer 2 Report

The manuscript has been appropriately revised, and the responses are reasonable. In my opinion, proper reference(s) should be given for the coherence length (line 285). The manuscript is now acceptable for publication.